

# Monthly streamflow forecasting at varying spatial scales in the Rhine basin

Simon Schick[1,2], Ole Rössler[1,2], and Rolf Weingartner[1,2]

[1]Institute of Geography, University of Bern, Bern, Switzerland
[2]Oeschger Centre for Climate Change Research, University of Bern, Bern, Switzerland

*Correspondence to:* Simon Schick (simon.schick@giub.unibe.ch)

**Abstract.** Model output statistics (MOS) methods empirically relate an environmental variable of interest to predictions from general circulation models (GCMs). This variable often belongs to a spatial scale not resolved by the GCM. Here, using the linear model fitted by least squares, we regress monthly mean streamflow of the Rhine River at Lobith and Basel against seasonal predictions of precipitation, surface air temperature, and runoff from the European Centre for Medium-Range Weather Forecasts. To address potential effects of a scale mismatch between the GCM's horizontal grid resolution and the hydrological application, the MOS method is further tested with an experiment conducted at the subcatchment scale. This experiment applies the MOS method to 133 additional gauging stations located within the Rhine basin and combines the forecasts from the subcatchments to predict streamflow at Lobith and Basel. In so doing, the MOS method is tested for catchments areas covering four orders of magnitude. Using data from the period 1981-2011, the results show that skill, with respect to climatology, is restricted to the first month ahead. This result holds for both the predictor combination that mimics the initial conditions and the predictor combinations that additionally include the dynamical seasonal predictions. The latter, however, reduces the mean absolute error of the former in the range of 5 to 11 percent, which is consistently reproduced at the subcatchment scale. The results further indicate that bias corrected runoff from the H-TESSEL land surface model is an interesting option when it comes to seasonal streamflow forecasting in large river basins.

## 1 Introduction

Environmental forecasting at the subseasonal to seasonal time scale promises a basis for planning in e.g. energy production, agriculture, shipping, or water resources management. While the uncertainties of these forecasts are inherently large, they can be reduced when the quantity of interest is controlled by slowly-varying and predictable phenomena, of which the El Niño-Southern Oscillation might be the most prominent one (National Academies, 2016).

In case of streamflow forecasting the ESP-revESP experiment proposed by Wood and Lettenmaier (2008) provides a methodological framework to disentangle forecast uncertainty with respect to the initial conditions and the meteorological forcings. Being a retrospective simulation, the experiment consists of model runs where the initial conditions are assumed to be known and the meteorological forcing series are randomly drawn (ESP, Ensemble Streamflow Prediction) and vice versa (revESP,





reverse Ensemble Streamflow Prediction). In this context the initial conditions refer to the spatial distribution, volume, and phase of water in the catchment at the date of prediction.

The framework allows for the estimation of the time range at which the initial conditions control the generation of streamflow: When the prediction error of the ESP simulation exceeds that of the revESP simulation, the meteorological forcings start

to dominate the streamflow generation. Similarly, when the prediction error of the ESP simulation approaches the prediction error of the climatology (i.e. average streamflow used as naive prediction strategy), the initial conditions no longer control the streamflow generation.

In both cases this time range depends on the interplay between climatological features (e.g. transitions between wet and dry or cold and warm seasons) and catchment specific hydrological storages (e.g. surface water bodies, soils, aquifers, and

10 snow) and can vary from zero up to several months (van Dijk et al., 2013; Shukla et al., 2013; Yossef et al., 2013). Indeed, this source of predictability is the rationale behind the application of the ESP approach in operational forecast settings, and it can be further exploited by conditioning on climate precursors (e.g. Beckers et al., 2016).

An emerging option for seasonal streamflow forecasting is the integration of seasonal predictions from coupled atmosphere-ocean-land general circulation models (Yuan et al., 2015b). Predictions from a general circulation model (GCM) can be used

threefold to the aim of streamflow forecasting by

1. forcing a hydrological model with the predicted evolution of the atmosphere;

2. employing runoff simulated by the land surface model, eventually in combination with a routing model;

3. using the predicted states of the atmosphere, ocean, or land surface in a perfect prognosis or model output statistics context with the streamflow as the predictand.

The first approach requires a calibrated hydrological model for the region of interest. In order to correct a potential bias and to match the spatial and temporal resolution of the hydrological model, it further involves a postprocessing of the atmospheric fields. A postprocessing also might be applied to the streamflow forecasts to account for deficiencies of the hydrological model. See e.g. Yuan et al. (2015a) or Bennett et al. (2016) for recent implementations of such a model chain.

In the second approach the land surface model takes the hydrological model's place with the difference that the atmosphere

and land surface are fully coupled. Since land surface components of coupled GCMs often represent groundwater dynamics and the river routing in a simplified way (Clark et al., 2015), the simulated runoff might be fed to a routing model as e.g. in Pappenberger et al. (2010). To the best of our knowledge, this approach has not yet been tested with a specific focus on the seasonal time scale.

The third approach deals with developing an empirical prediction rule for streamflow. If the model building procedure is

30 based on observations only, the approach is commonly referred to as perfect prognosis (PP). On the other hand, the model might be built using the hindcast archive of a particular GCM (model output statistics, MOS). In both cases the final prediction rule is applied to the actual GCM outcome to forecast the quantity of interest. Therefore, MOS methods require the presence of a hindcast archive of the involved GCM, but can take systematic errors of the GCM into account (Brunet et al., 1988).





Only a few studies map GCM output to streamflow with PP or MOS methods, including multiple linear regression (Marcos et al., 2017), principal components regression and canonical correlation analysis (Foster and Uvo, 2010; Sahu et al., 2016), artificial neural networks (Humphrey et al., 2016), and an ensemble of generalized linear models, locally weighted polynomial regression, and k-nearest-neighbour prediction rules (Chowdhury and Sharma, 2009). By far the most selected predictor is

catchment area precipitation, but depending on the study region also surface air temperature, sea surface temperature, or wind velocity are used. Whatever the selected predictors, PP and MOS methods often conduct the mapping across spatial scales. For example, if the catchment of interest falls below the grid scale of the GCM, PP and MOS methods implicitly perform a downscaling step. If the catchment covers several grid points, the method implicitly performs an upscaling.

The present study aims to take up this scale bridging and to test a MOS-based approach for seasonal streamflow forecasting

and a range of catchment areas. To analyse the limits of predictability and to aid interpretation, we first adapt the ESP-revESP framework to the context of regression by defining predictor combinations that conceptually correspond to the ESP and revESP simulations. Next, seasonal predictions of precipitation, surface air temperature, and runoff from the European Centre for Medium-Range Weather Forecasts (ECMWF) enter the regression model and the resulting forecast skill is estimated with respect to the ESP-like regression model.

The variation of the catchment area borrows from the concept of the 'working scale' (Blöschl and Sivapalan, 1995): Given a particular target catchment, the regression models are applied at the catchment scale as well as two levels of subcatchment scales. In case of the subcatchments, the resulting forecasts are combined in order to get a forecast at the outlet of the target catchment. By validating the combined forecasts of the subcatchments at the main outlet, any differences in the forecast quality can be attributed to the working scales.

This experiment is conducted for the Rhine River at Lobith and Basel in Western Europe. In general the current quality of seasonal climate predictions is classified to be low for this region (Kim et al., 2012; Doblas-Reyes et al., 2013). Streamflow hindcast experiments with dynamical seasonal climate predictions, however, indicate skill beyond the lead time of traditional weather forecasts: Concerning catchments of the Alpine and High Rhine, Orth and Seneviratne (2013) estimate the skillful lead time for daily mean streamflow to lie between one and two weeks, which increases to about one month when focusing on low

flows (Fundel et al., 2013; Jörg-Hess et al., 2015). Also for daily low flow Demirel et al. (2015) report for the Moselle River a sharp decrease in skill after 30 days. For a set of French catchments Crochemore et al. (2016) show that weekly streamflow forecasts are improved for lead times up to about one month when using postprocessed seasonal precipitation predictions. Singla et al. (2012) advance spring mean streamflow forecasts for the French part of the Rhine basin with seasonal predictions of precipitation and surface air temperature.

The above studies show that in case of the Rhine basin current model chains skillfully forecast daily mean streamflows approximately two weeks ahead. When considering low flows only, these two weeks extend to about one month, and by reducing the forecasts temporal resolution even longer forecast ranges seem to be feasible. As a compromise between skillful lead time and temporal resolution, we decide to focus on monthly mean streamflow at lead times of zero, one, and two months. Here, zero lead time refers to forecasting the next month, while e.g. the one month lead time denotes a temporal gap of one



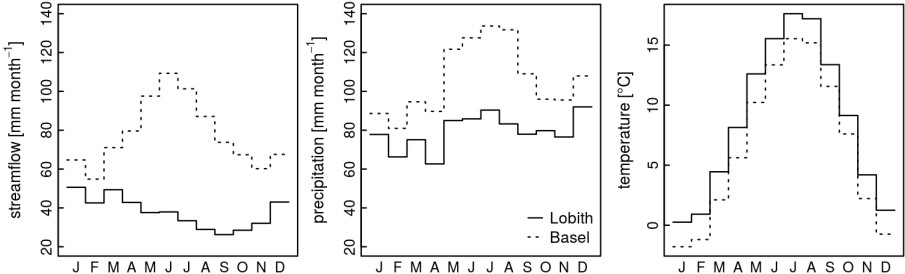

**Figure 1.** Monthly area averages of streamflow, precipitation, and surface air temperature for the Rhine at Lobith and Basel with respect to the period 1981-2011 (GRDC, 2016; E-OBS, 2016).

month between the release of a forecast and its time of validity. Strictly speaking, the present study deals with hindcasts or retrospective forecasts. However, for the sake of readability we use the terms forecast, hindcast, and prediction interchangeably.

Below, Sect. 2 introduces the study region, Sect. 3 describes the data set, Sect. 4 exposes the methodology in more detail, and in Sect. 5 and 6 the results are presented and discussed, respectively.

## 2 Study region

The Rhine River is situated in Western Europe and discharges into the North Sea; in the south its basin is defined by the Alps. About 58 million people use the Rhine water for the purpose of navigation, hydro power, industry, agriculture, drinking water supply, and leisure (ICPR, 2009). The present study focuses on two gauging stations: The first is located in Lobith near the Dutch-German border, the second in Basel in the tri-border region of France, Germany, and Switzerland.

Table 1 lists some geographical attributes. The Rhine at Basel covers an area of approximately one fifth of the Rhine at Lobith whereas the mean elevation halves when going from Basel to Lobith. The negative minimum elevation of the Rhine at Lobith is due to a coal mine. Dominant land use classes are farmed areas and forests, but the Rhine at Basel proportionately includes more grass land, wasteland, surface water, and glacier.

Concerning the climatology of the period 1981-2011 (Fig. 1), we observe that streamflow peaks at Lobith in winter and at Basel in early summer. Streamflow at Basel is dominated by snow accumulation in winter, subsequent snow melting in spring, and high precipitation in summer. At Lobith precipitation exhibits less variability and higher surface air temperature intensifies evaporation. Based on recent climate projections, it is expected that streamflow in the Rhine basin is going to increase in winter, to decrease in summer, and to slightly decrease in its annual mean in the last third of the 21th century (Bosshard et al., 2014).

## 3 Data

Observations of river streamflow and gridded runoff, precipitation, and surface air temperature of the period 1981-2011 in daily resolution constitute the data set. Throughout the study gridded quantities get aggregated to (sub)catchment area averages.



**Table 1.** Geography of the Rhine River at Basel and Lobith according to CORINE (2013), EU-DEM (2013), and GRDC (2016).

|  | Lobith | Basel |
| --- | --- | --- |
| area (km$^2$) | 159700 | 36000 |
| gauging station (m a. s.) | 20 | 250 |
| elevation min (m a. s.) | -230 | 250 |
| elevation max (m a. s.) | 4060 | 4060 |
| elevation mean (m a. s.) | 490 | 1050 |
| farmed area (%) | 47.7 | 36.8 |
| forest (%) | 35.8 | 31.6 |
| grass land (%) | 3.4 | 11.4 |
| urban area (%) | 9.6 | 7.0 |
| wasteland (%) | 1.8 | 8.2 |
| surface water (%) | 1.4 | 4.0 |
| glacier (%) | 0.3 | 1.0 |

The streamflow observations consist of a set of 135 time series in m$^3$ s$^{-1}$. These series as well as the corresponding catchment boundaries are provided by several public authorities and the Global Runoff Data Centre (GRDC (2016); see also Sect. 9), and belong to catchments with nearly natural to heavily regulated streamflow.

The ENSEMBLES gridded observational data set in Europe (E-OBS, version 14.0) provides precipitation and surface air temperature on the 0.25° regular grid (Haylock et al., 2008; E-OBS, 2016). These fields base upon the interpolation of station data and are subject to inhomogeneities and biases. However, a comparison against meteorological fields derived from denser station networks attests a high correlation (Hofstra et al., 2009).

Precipitation, surface air temperature, and runoff from ECMWF's seasonal forecast system 4 (S4) archive are on a 0.75° regular grid. This hindcast set consists of 15 members of which we take the ensemble mean. Runs of the coupled atmosphere-ocean-land model are initialised on the first day of each month with a lead time of seven months. Up to 2010, initial conditions are out of ERA Interim, and the year 2011 is based on the operational analysis.

The atmospheric model (IFS cycle 36r4) consists of 91 vertical levels with the top level at 0.01 hPa in the mesosphere. The horizontal resolution is truncated at TL255 and the temporal discretisation equals 45 min. The NEMO ocean model has 42 levels with a horizontal resolution of about 1°. Sea ice is considered by using its actual extent from the analysis and relaxing it towards the climatology of the past five years (Molteni et al., 2011).

The H-TESSEL land surface model implements four soil layers with an additional snow layer on the top. Interception, infiltration, surface runoff, and evapotranspiration are dealt with by dynamically separating a grid cell in to fractions of bare ground, low and high vegetation, intercepted water, and shaded and exposed snow. In contrast, the soil properties of a particular layer are uniformly distributed within one grid cell. Vertical water movement in the soil follows Richards's equation with an





additional sink term to allow for water uptake by plants. Runoff per grid cell then finally equals the sum of surface runoff and open drainage at the soil bottom (ECMWF, 2016).

In case of the Rhine basin an E-OBS tile in the above configuration approximately covers an area of $500\ \mathrm{km}^2$, and an S4 tile an area of about $4500\ \mathrm{km}^2$.

# 4 Method

The following subsections outline the experiment, which is individually conducted for both the Rhine at Lobith and Basel. Section 4.1 first details the predictor combinations and the regression strategy. Section 4.2 introduces the variation of the catchment area and Sect. 4.3 illustrates the validation of the resulting hindcasts.

## 4.1 Model building

Let $y_{i,j}$ denote observations of mean streamflow at a specific gauging site in $\mathrm{m^3\,s^{-1}}$ for $j = 30, 60, 90$ d, starting the first day of each calendar month $i = 1, ..., 12$ in the period 1981-2011. Henceforth $y_{i,j}$ is the predictand.

### 4.1.1 Predictor combinations

The set of predictors consists of variables that either precede or succeed the date of prediction, i.e the first day of month $i$ (Tab. 2). The first model refRun (reference run) is aimed to estimate how well the regression works given the best available
input data. The second and third combinations imitate the ESP and revESP simulations. The ESP-revESP framework thus is mimicked by constraining the model to observed precipitation and temperature either prior to or following the date of prediction.

The S4* combinations actually constitute the MOS method and consider the seasonal predictions out of the S4 hindcast archive, where we use the asterisk as wildcard to refer to any of the S4P, S4T, S4PT, and S4Q models. The S4P and S4T
models are used to separate the forecast quality with respect to precipitation and temperature. The S4Q model is tested as H-TESSEL does not implement any groundwater dynamics and preceding precipitation and temperature might tap this source of predictability. Let aside the S4Q model, the preceding and subsequent predictors conceptually approximate the initial conditions and the meteorological forcings, respectively.

### 4.1.2 Regression

For a particular predictor combination and $y_{i,j}$ we first apply a correlation screening to select the optimal aggregation time $a_{i,j}$ for each predictor:

$$a_{i,j} = argmax_k \mid cor(y_{i,j}, x_{i,k}) \mid \tag{1}$$





**Table 2.** Predictor combinations consisting of (with respect to the date of prediction) preceding and subsequent precipitation ($p$), surface air temperature ($t$), and runoff ($q$); the numerical values are either out of E-OBS or the S4 hindcast archive.

|  | preceding | | subsequent | | |
| --- | --- | --- | --- | --- | --- |
| model | $p^{\mathrm{pre}}$ | $t^{\mathrm{pre}}$ | $p^{\mathrm{sub}}$ | $t^{\mathrm{sub}}$ | $q^{\mathrm{sub}}$ |
| refRun | E-OBS | E-OBS | E-OBS | E-OBS | - |
| ESP | E-OBS | E-OBS | - | - | - |
| revESP | - | - | E-OBS | E-OBS | - |
| S4P | E-OBS | E-OBS | S4 | - | - |
| S4T | E-OBS | E-OBS | - | S4 | - |
| S4PT | E-OBS | E-OBS | S4 | S4 | - |
| S4Q | E-OBS | E-OBS | - | - | S4 |

where $x_{i,k}$ is one of the predictors from Tab. 2 and $k = -10, -20, ..., -720$ d in case of $p^{\mathrm{pre}}$ and $t^{\mathrm{pre}}$ (backward in time relative to the date of prediction) and $k = 5, 10, ..., j$ d in case of $p^{\mathrm{sub}}$, $t^{\mathrm{sub}}$, and $q^{\mathrm{sub}}$ (forward in time relative to the date of prediction). The limit of 720 d is chosen since larger values rarely get selected.

The ordinary least squares hyperplane is then used for prediction without any transformation, basis expansion, or interaction. However, model variance can be an issue: Specifically for the ESP model from Tab. 2 we expect the signal-to-noise ratio to be low in most of the seasons. In combination with the moderate sample size $n = 31$ for model fitting, perturbations in the training set can lead to large changes in the predictors time lengths $a_{i,j}$ and regression coefficients. In order to reduce model variance, we draw 100 non-parametric bootstrap replicates of the training set, fit the model to these replicates, and combine the predictions by unweighted averaging (Breiman, 1996; Schick et al., 2016).

### 4.1.3 Cross-validation

Each year with a buffer of two years (i.e. the two preceding and subsequent years) is left out and the regression outlined in Sect. 4.1.2 is applied to the remaining years. The fitted models then predict the central left-out years. Buffering is used to avoid artificial forecast quality due to hydrometeorological persistence (Michaelsen, 1987).

### 4.1.4 Lead time

Lead time is introduced by integrating the predicted $\hat{y}_{i,j}$ in time and taking differences with respect to $j$. For example monthly mean streamflow $z_i$ in July ($i = 7$) is predicted with a lead time of one month according to

$$\hat{z}_7 = (\hat{y}_{6,60} \cdot (30 + 31) \cdot s - \hat{y}_{6,30} \cdot 30 \cdot s)/(31 \cdot s) \tag{2}$$





**Table 3.** Subcatchment division of the Rhine at Lobith and Basel. The median area covers four orders of magnitude.

|  | number of subcatchments | area km$^2$ | | |
|---|---|---|---|---|
|  |  | min | median | max |
| Lobith level 1 | 1 | - | 159700 | - |
| Lobith level 2 | 5 | 19690 | 33220 | 43550 |
| Lobith level 3 | 12 | 8284 | 13040 | 17610 |
| Basel level 1 | 1 | - | 36000 | - |
| Basel level 2 | 10 | 1871 | 2946 | 6346 |
| Basel level 3 | 124 | 6 | 187 | 2654 |

where $s = 24 \cdot 60 \cdot 60$ s equals the number of seconds of one day and both $\hat{y}$ and $\hat{z}$ have unit m$^3$ s$^{-1}$. For zero lead time, we set $\hat{z}_i = \hat{y}_{i,30}$. Please note that the year 1981 needs to be dropped from the validation (Sect. 4.3) since the length of the streamflow series prevents to forecast e.g. January 1981 with a lead time of one month.

## 4.2 Spatial levels

Contrasting the forecast quality of a given model for individual catchments separated in space inevitably implies a large number of factors, e.g. the geographic location (and thus the involved GCM grid points), the orography, or the degree to which streamflow is regulated. In order to hold these factors whilst screening through a range of catchment areas, we propose to vary the working scale within a particular target catchment.

Following this line of argumentation we apply the model building procedure from Sect. 4.1 to three distinct sets of subcatch-
ments, which we term 'spatial levels' (Tab. 3). Spatial level 1 simply consists of the target catchment itself, i.e. the Rhine at Lobith and Basel. At spatial levels 2 and 3 we take additional gauging stations from within the Rhine basin, which naturally divide the basin into subcatchments.

For these subcatchments we have streamflow observations belonging to the entire upstream area, but not the actual subcatchment area itself. To arrive at an estimate of the water volume generated by the subcatchment, we equate the predictand $y_{i,j}$ to
the difference of outflow and inflow of that subcatchment. For a particular date of prediction and spatial level, the sum of the resulting subcatchment forecasts $\hat{z}_i$ then constitutes the final forecast for the Rhine at Lobith and Basel, respectively.

A drawback of this procedure is that we ignore the water travel time: First when taking the differences of outflows and inflows and second when summing up the subcatchment forecasts. While the former increases the observational noise, the latter does not affect the regression itself, but adds a noise term to the final forecast at Lobith and Basel. As the statistical
properties of the noise introduced by the water travel time is unknown, we only can argue that the results below provide a lower bound of the forecast quality due to this methodological constraint.





### 4.3 Validation

The forecast quality of the regression models is analysed with the pairs of cross-validated monthly mean streamflow forecasts and observations $(\hat{z}, z)$. These series cover the period 1982-2011 and have a sample size of $n = 360$. The first validation steps focus on the forecasts at Lobith and Basel and thus consider the sum of the subcatchments forecasts $\hat{z}$ per spatial level. The forecasts in the subcatchments itself are addressed in Sect. 4.3.4.

#### 4.3.1 Benchmarks

Climatology and runoff simulated by H-TESSEL serve as benchmarks. The monthly climatology is estimated with the arithmetic mean from the daily streamflow observations. The monthly basin averages of H-TESSEL get post-calibrated via linear regression against the monthly mean streamflow observations at Lobith and Basel, respectively. For both benchmarks the cross-validation scheme from Sect. 4.1.3 is applied.

#### 4.3.2 Taylor diagram

Taylor diagrams (Jolliffe and Stephenson, 2012) are employed to get a global overview. For a particular model, let $\rho$ be the Pearson correlation coefficient of the forecasts $\hat{z}$ and the corresponding observations $z$. The plotting position of the model has a distance from the origin equal to the standard deviation of $\hat{z}$ and is located on the line having an angle of incline $\phi = arccos(\rho)$. The plotting position of the observations $z$ has a distance from the origin equal to the standard deviation of $z$ and is located on the abscissa. The distance between the two plotting positions equals the root mean squared error with the unconditional bias $E(\hat{Z} - Z)$ removed.

#### 4.3.3 Mean absolute error

The statistical significance of the difference in forecast accuracy between the ESP and a S4* model is tested in terms of the mean absolute error (MAE). As it turns out, the paired differences of absolute errors for a given lead time and spatial level

$$d = \mid \hat{z}^{\text{ESP}} - z \mid - \mid \hat{z}^{\text{S4*}} - z \mid \tag{3}$$

no longer exhibit serial correlation and approximately follow a Gaussian distribution. Using the mean difference $\bar{d}$, we then report the p-values of the two-sided t-test with null hypothesis $\bar{d} = 0$ and alternative hypothesis $\bar{d} \neq 0$. The sample autocorrelation functions and quantile plots against the Gaussian distribution of $d$ for zero lead time are included in the additional materials (Sect. 10).

To evaluate whether a particular model $m$ has skill with respect to a reference model $r$ the MAE ratio

$$s = 1 - \frac{\text{MAE}_m}{\text{MAE}_r} \tag{4}$$

is employed. For example, $m$ could be a S4* model and $r$ the ESP model. $s = 0.1$ means that the model $m$ lowers the MAE of model $r$ by 10 %.





### 4.3.4 Subcatchments

To help in the interpretation of the forecast quality of the MOS method regarding the spatial levels at Lobith and Basel, we finally have a look at the subcatchments itself, which are up to now only implicitly addressed. In a qualitative manner we plot the MAE skill score (Eq. 4) of the S4* and ESP models in space as well as against the subcatchment area, the median of the terrain roughness, the MAE skill score of the revESP with the ESP model as reference, and the MAE skill score of the refRun model with the climatology as reference.

The terrain roughness is included since the atmospheric flow in complex terrain is challenging to simulate and atmospheric GCMs need to filter the topography according to their spatial resolution (Maraun and Widmann, 2015; Torma et al., 2015). The terrain roughness is defined as the difference of the maximum and minimum elevation value within a 3 times 3 pixel window (Wilson et al., 2007). It is derived here from the digital elevation model EU-DEM (2013), which has a horizontal resolution of 25 m.

## 5 Results

### 5.1 Taylor diagram

Figure 2 shows the Taylor diagrams for Lobith and Basel to get a global overview regarding the lead times, predictor combinations, and spatial levels. Accurate forecasts reproduce the standard deviation of the observations (thus lie on the circle with radius equal to the the standard deviation of the observations), and also exhibit high correlation (so travel on this circle towards the observations on the abscissa). At a first glimpse the spatial levels do not introduce clear differences and most of the models mass at the same spots.

The benchmark climatology is outperformed at zero lead time by all models. At longer lead times the revESP model pops up besides the refRun model and the remaining models approach climatology. H-TESSEL stays close to the regression models and tends to score a higher correlation in case of Lobith, but not Basel. For the refRun model we note a correlation of about 0.9 independently of the lead time while the observations variability generally is underestimated.

For Lobith and zero lead time we observe an elongated cluster, which comprises all models but the climatology and the refRun model. Some models score a higher correlation – zooming in would reveal that these are the S4P, S4PT, and S4Q models with H-TESSEL standing at the forefront. In the following we focus on the forecasts with zero lead time since at longer lead times we virtually do not have any improvements relative to the climatology.

### 5.2 Mean absolute error

Table 4 reports the mean absolute error (MAE) at zero lead time. Reading Tab. 4 along the rows reveals a more or less consistent pattern: The refRun model approximately halves the MAE of the climatology; differences between the ESP, revESP, and S4T models are small; compared to the ESP model, the S4P, S4PT, and S4Q models lower the MAE by about $40\,\mathrm{m^3\,s^{-1}}$ for Lobith and by about $15\,\mathrm{m^3\,s^{-1}}$ for Basel; and H-TESSEL outperforms the S4* models in case of Lobith, but not Basel. When reading





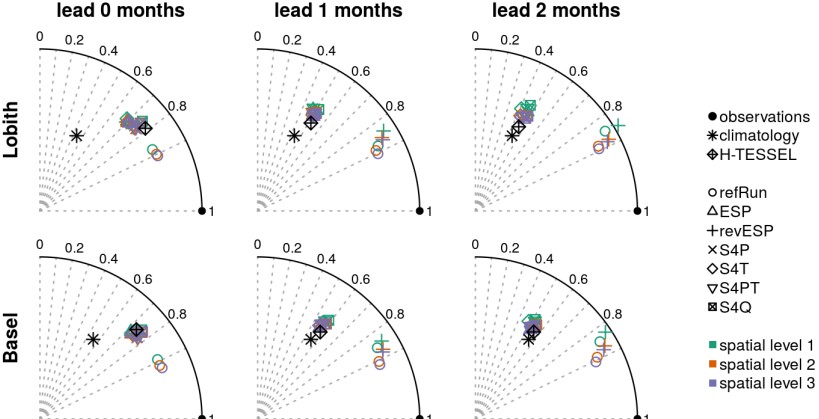

**Figure 2.** Taylor diagrams for the benchmarks climatology and H-TESSEL and the predictor combinations from Tab. 2 at Lobith (top row) and Basel (bottom row).

Tab. 4 along the columns, we generally note at Lobith a decreasing MAE when going from spatial level 1 to spatial level 3. In case of Basel, the MAE remains more or less constant except for the refRun model.

Focusing on the MOS method, Tab. 5 contains the p-values for the null hypothesis 'the ESP and S4* models score an equal mean absolute error'. Apart from the S4T model the results among the spatial levels agree. While at Lobith the null hypothesis

for the S4T model should not be rejected, at Basel one might do so.

Table 6 shows the corresponding MAE skill score (Eq. 4) using the ESP model as reference. The S4P, S4PT, and S4Q models score an error reduction ranging from 5 to 11 %. In case of the S4T model an error reduction is not existent (Lobith) or small (Basel), supporting the high p-values from Tab. 5. The MAE reduction generally tends to increase along the spatial levels, however, on a rather low level.

In order to reduce the number of models, we drop the S4P, S4T, and S4Q models and focus in the next section on the S4PT model. Temperature is retained as predictor because the S4T model might not be rejected at Basel (Tab. 5). Among the similar performing S4P, S4PT, and S4Q models, the S4PT model is selected for ease of interpretation as the refRun, ESP, revESP, and S4PT models share the same predictors. For the sake of completeness Fig. 3 and 4 below are included in the additional materials for the dropped S4* models (Sect. 10).

## 5.3  Subcatchments

Figure 3 depicts the MAE skill score (Eq. 4) for the S4PT model relative to the ESP model for each subcatchment at zero lead time. If the MAE difference does not exhibit a p-value smaller than 0.05 (Eq. 3), the subcatchment is coloured in white. We observe that the MAE skill score takes values in the range of -0.06 to 0.12 and both the lowest and highest scores occur at Basel and spatial level 3. Negative scores can only be found at Basel and spatial level 3, and positive skill tends to cluster in

space.





**Table 4.** Mean absolute error at zero lead time of the benchmarks climatology and H-TESSEL and the predictor combinations from Tab. 2, rounded to integers. All values have unit $\mathrm{m^3\,s^{-1}}$; $n = 360$.

|  | climatology | H-TESSEL | refRun | ESP | revESP | S4P | S4T | S4PT | S4Q |
|---|---|---|---|---|---|---|---|---|---|
| Lobith level 1 | 633 | 419 | 334 | 499 | 499 | 460 | 506 | 464 | 446 |
| Lobith level 2 |  |  | 299 | 484 | 497 | 440 | 484 | 445 | 442 |
| Lobith level 3 |  |  | 288 | 480 | 495 | 437 | 479 | 442 | 439 |
| Basel level 1 | 239 | 191 | 130 | 201 | 194 | 188 | 195 | 187 | 190 |
| Basel level 2 |  |  | 118 | 199 | 192 | 185 | 194 | 184 | 187 |
| Basel level 3 |  |  | 113 | 199 | 193 | 184 | 195 | 183 | 187 |

**Table 5.** p-values for the null hypothesis 'the ESP and S4* models score an equal mean absolute error' at zero lead time; $n = 360$.

|  | S4P | S4T | S4PT | S4Q |
|---|---|---|---|---|
| Lobith level 1 | <0.01 | 0.32 | <0.01 | <0.01 |
| Lobith level 2 | <0.01 | 0.95 | <0.01 | <0.01 |
| Lobith level 3 | <0.01 | 0.89 | <0.01 | <0.01 |
| Basel level 1 | <0.01 | 0.03 | <0.01 | 0.01 |
| Basel level 2 | <0.01 | 0.05 | <0.01 | <0.01 |
| Basel level 3 | <0.01 | 0.08 | <0.01 | <0.01 |

The same skill scores from Fig. 3 are contrasted in Fig. 4 with the subcatchment area, the median of the terrain roughness, the MAE skill score of the revESP model relative to the ESP model, and the MAE skill score of the refRun model relative to the climatology. While the first two attributes concern the geography of the subcatchment, the third attribute indicates the relevance of the initial conditions for the subsequent generation of streamflow. The fourth attribute shows how well the S4PT model performs relative to the climatology as benchmark, when it has access to the best available input data.

In addition to the MAE skill scores of the subcatchments, the horizontal lines in Fig. 4 depict the MAE skill scores for each spatial level at Lobith and Basel (i.e. the values from Tab. 6). If the MAE difference does not exhibit a p-value smaller than 0.05 (Eq. 3), the symbol is drawn with a reduced size.

The resulting patterns suggest that positive skill does not depend on the subcatchment area. On the other hand, a low terrain roughness and a weak relevance of the initial conditions seem to favour positive skill. The last row finally indicates that positive skill is restricted to subcatchments where the refRun model outperforms climatology. Roughly, a hypothetical linear relationship seems to strengthen from the top to the bottom plots.





**Table 6.** MAE skill score of the S4* models relative to the ESP model (Eq. 4), expressed in percent; $n = 360$.

|  | S4P | S4T | S4PT | S4Q |
|---|---|---|---|---|
| Lobith level 1 | 8 | -1 | 7 | 11 |
| Lobith level 2 | 9 | 0 | 8 | 9 |
| Lobith level 3 | 9 | 0 | 8 | 9 |
| Basel level 1 | 6 | 3 | 7 | 5 |
| Basel level 2 | 7 | 2 | 7 | 6 |
| Basel level 3 | 8 | 2 | 8 | 6 |

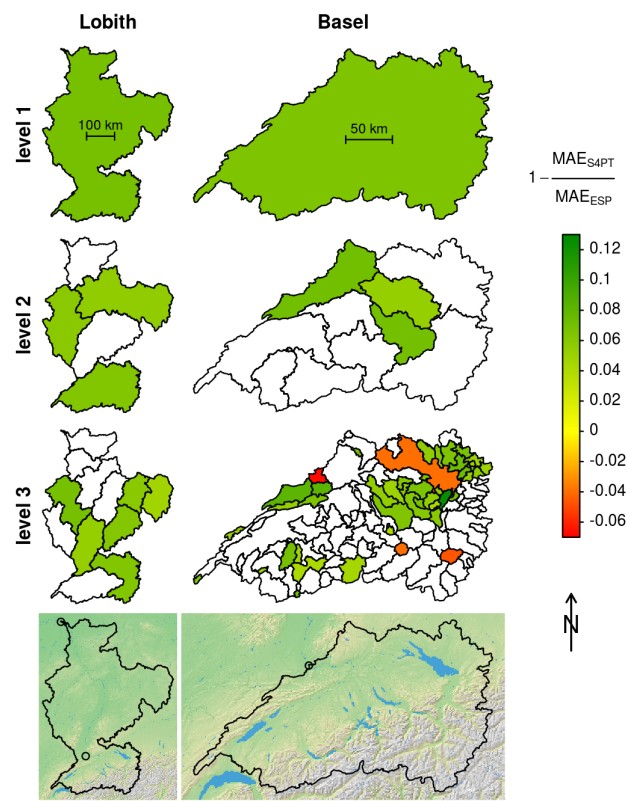

**Figure 3.** MAE skill score of the S4PT model with respect to the ESP model for each subcatchment and zero lead time. Subcatchments are coloured only when the p-value for the null hypothesis 'the ESP and S4PT models score an equal mean absolute error' is smaller than 0.05. In the bottom maps the main outlets at Lobith and Basel are marked with a black circle and open water surfaces are coloured in blue (CORINE, 2013; EU-DEM, 2013).





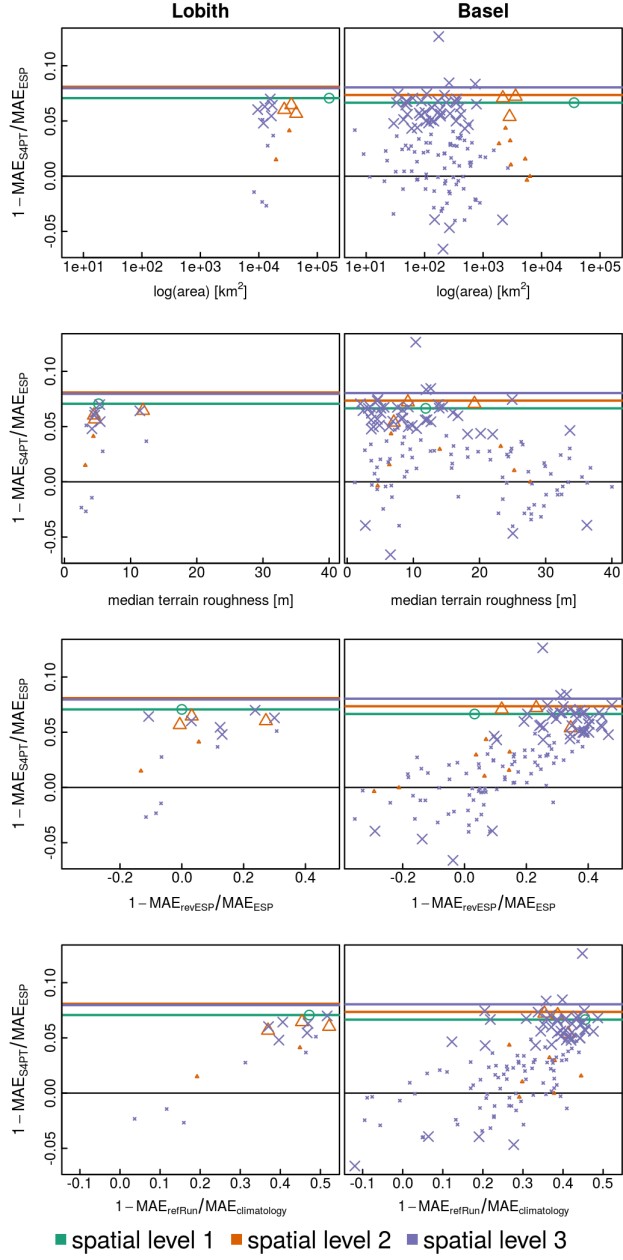

**Figure 4.** MAE skill score of the S4PT model with respect to the ESP model for each subcatchment and zero lead time, plotted against subcatchment attributes (see Sect. 4.3.4 for details). Lines indicate the corresponding skill per spatial level at Lobith and Basel. Large symbols note a p-value smaller than 0.05 for the null hypothesis 'the ESP and S4PT models score an equal mean absolute error'.



## 6   Discussion

The following discussion is valid only for predicting monthly mean streamflow throughout the complete calendar year. An evaluation of the forecast quality with respect to particular calendar months goes beyond the scope of the study.

### 6.1   Model building

The refRun model, which has access to the best available input data, ends up with a correlation of about 0.9 for all lead times, spatial levels, and both Lobith and Basel (Fig. 2). Part of this correlation is also the annual cycle (Fig. 1), which already leads to a correlation of about 0.5 when using the climatology as prediction rule. The forecasts from the refRun model do not fully reproduce the observations variance, what might be improved with a transformation of the predictand (Wang et al., 2012). This option preferably should be tested in a future study with a small number of catchments and longer time series.

### 6.2   Spatial levels

Besides the ignorance of the water travel time (Sect. 4.2), the spatial levels basically can degrade the forecast quality in three ways. For a particular subcatchment,

1.  the assumption of a linear relationship between the predictors and the predictand might not be valid;

2.  the present variables precipitation, surface air temperature, and runoff simply do not contain any relevant information
(for example due to heavily regulated streamflow);

3.  the aggregation of the E-OBS and S4 fields at the catchment scale is not the appropriate spatial resolution (e.g. large scale grid averages cancel any spatial variability, and for catchment areas below the grid scale a grid point does not necessarily contain information valid at the local scale).

Despite these three sources of uncertainty and the ignorance of the water travel time, we only observe a small gradual
improvement of the forecast accuracy along the spatial levels (Tab. 4). While this result does not allow to relate the forecast accuracy to these uncertainties, it supports at least the robustness of the estimated MAE skill score for the forecasts at Lobith and Basel (Tab. 4): Applying the regression models at spatial levels 2 and 3 virtually does not include streamflow information at Lobith and Basel (with the exception of the subcatchments that include these gauges), so artificial skill can hardly be an issue.

### 6.3   ESP-revESP

In Yossef et al. (2013) the ESP-revESP framework is applied to the worlds largest river basins using the global hydrological model PCRaster Global Water Balance (PCR-GLOBWB). Considering all calendar months and the Rhine at Lobith, the ESP simulation outperforms the climatology only at zero lead time; the revESP simulation is outperformed at zero lead time by both the ESP simulation and climatology; and at longer lead times the revESP simulation clearly outperforms both the ESP




simulation and climatology. Therefore, the results of Yossef et al. (2013) and those of the present study are partly in line – initial conditions are skillful at zero lead time, but for unknown reasons a clear difference between the ESP and revESP model at zero lead time does not exist in our results.

## 6.4 MOS method

5 In case of the monthly mean streamflow forecasts at zero lead time, the MOS method based on precipitation or runoff provides a smaller mean absolute error than the ESP model (Tab. 6). Here, it must be stressed that for the present regression strategy subsequent temperature often is a weak predictor (not shown). Thus, a possible rejection of the S4T model does not allow any inference about the forecast quality of surface air temperature itself.

While the variation of the MAE skill score along the spatial levels is small (Tab. 6), the skill in the subcatchments itself varies 10 considerably (Fig. 3 and 4). The integration of the seasonal predictions from S4 frequently leads to negative MAE skill scores. Negative scores arise when the model catches spurious relationships, which subsequently get penalised during cross-validation. These negative scores need to be compensated in order to outperform the ESP model at Lobith and Basel.

Figure 4 indicates that the subcatchment area most likely is not relevant to score positive skill. Rather the S4PT model outperforms the ESP model in subcatchments where the terrain roughness and the relevance of the initial conditions is low. 15 However, the terrain roughness and the relevance of the initial conditions are not independent attributes: Fig. 3 shows that for small subcatchments in the alpine region positive skill is sparely present (spatial levels 2 and 3 at Basel). These subcatchments generally exhibit a high terrain roughness as well as a high relevance of the initial conditions due to snow accumulation in winter and subsequent melting in spring and summer.

Somewhat trivial, Fig. 4 also shows that skill of the S4PT model is restricted to subcatchments where the refRun model 20 outperforms climatology. If the refRun model downgrades to the climatology, precipitation and temperature do not contain any relevant information to predict streamflow. Consequently also the dynamical seasonal predictions are, however accurate, useless.

## 6.5 H-TESSEL

An interesting result finally is the performance of H-TESSEL. Within ECMWF's seasonal forecasting system S4, H-TESSEL 25 is aimed to provide a lower boundary condition for the simulation of the atmosphere and consequently does neither implement streamflow routing nor ground water storage (ECMWF, 2016). According to Tab. 4 H-TESSEL in combination with a linear bias correction best translates the seasonal predictions in case of Lobith among the models that could be used in an operational forecast setting.

The S4Q model, which has access to the same input data and in addition conditions on preceding precipitation and temper- 30 ature, scores a lower forecast accuracy than H-TESSEL in case of Lobith (Tab. 4). This most likely is related to overfitting, which is not sufficiently smoothed by the model averaging (Sect. 4.1.2). The question remains whether a more advanced post-processing instead of the simple linear bias correction leads to further improvements, e.g. by conditioning on other variables or by using a river routing model.




## 7 Conclusions

The present study tests a model output statistics (MOS) method for monthly mean streamflow forecasts in the Rhine basin. The method relies on the linear regression model fitted by least squares and uses predictions of precipitation and surface air temperature from the seasonal forecast system S4 of the European Centre for Medium-Range Weather Forecasts. Observations of precipitation and surface air temperature prior to the date of prediction are employed to estimate the initial conditions. In addition, runoff simulated by the H-TESSEL land surface model is evaluated for its predictive power.

MOS methods often bridge the grid resolution of the general circulation model (GCM) and the spatial scale of the actual predictand. In order to estimate how the forecast quality depends on the catchment area, a hindcast experiment for the period 1981-2011 is conducted where the working scale is varied within the Rhine basin at Lobith and Basel. This variation is implemented by applying the MOS method to subcatchments and combining the resulting forecasts to predict streamflow at Lobith and Basel.

The monthly mean streamflow forecasts based on the initial conditions are skillful with respect to the climatology at zero lead time for both the Rhine at Lobith and Basel. The MOS method, which additionally has access to the dynamical seasonal predictions, further reduces the mean absolute error by about 5 to 11 % compared to the model that is constrained to the initial conditions. When the lead time is increased the forecasts virtually reduce to climatology. However, for a particular calendar month these findings can substantially deviate.

The above results hold for the entire range of tested subcatchment scales. Neither do effects of a scale mismatch between the GCM's horizontal grid resolution and the catchment area emerge, nor can a subcatchment scale be detected at which the MOS method clearly works best. Moreover, the results indicate that a skillful integration of the dynamical seasonal predictions requires catchments where the initial conditions are less relevant than the meteorological forcings.

The adaptation of the ESP-revESP framework proposed by Wood and Lettenmaier (2008) to the context of regression pays off in that it provides a reference model against which the MOS method can be tested. Clearly, when using regression the ESP-revESP framework does not provide the same insights as when using a hydrological simulation model, but nevertheless it can help in the interpretation of the results.

Given the present forecast quality of H-TESSEL in combination with a simple linear bias correction, we also conclude that runoff simulated by the land surface component of coupled GCMs is an interesting option when it comes to operational forecasting in large river basins. In addition, we think it could be interesting to establish such runoff simulations as a common benchmark in studies that use seasonal predictions from GCMs to forecast streamflow. Doing so could reveal where and why model chains, routing algorithms, MOS, and postprocessing techniques reduce uncertainties, and which hydrological processes can be implemented in a simplified manner to forecast at the seasonal time scale.

## 8 Code availability

The regression approach from Sect. 4.1.2 is compiled in an R package, which is included in the additional materials.





## 9 Data availability

E-OBS (2016), CORINE (2013), and EU-DEM (2013) are public data sets. Access to the ECMWF and GRDC archive must be requested. Data from the various public authorities as listed in the Acknowledgements is partly public.

## 10 Additional materials

Besides the R package and its vignette, the additional materials include Fig. 3 and Fig. 4 for the S4P, S4Q, and S4T models. Figure 5 shows per spatial level at Lobith and Basel and for each S4* model at zero months lead time: The sample autocorrelation function and quantile plots against the Gaussian distribution of the paired differences of absolute residuals with respect to the ESP model (Eq. 3), and scatterplots of predictions and observations.

*Acknowledgements.* Streamflow series and catchment boundaries are provided by the following authorities: State Institute for the Environ-
ment, Measurements and Conservation Baden Wuerttemberg; Bavarian Environmental Agency; State of Voralberg; Austrian Federal Ministry of Agriculture, Forestry, Environment and Water; and Swiss Federal Office for the Environment. Further we acknowledge the E-OBS data set from the EU-FP6 project ENSEMBLES (ensembles-eu.metoffice.com) and the data providers in the ECA&D project (www.ecad.eu) as well as the Copernicus data and information funded by the European Union (EU-DEM and CORINE). We also thank the Global Runoff Data Centre and the European Centre for Medium-Range Weather Forecasts for the access to the data archives. The study is funded by the Group
of Hydrology, which is part of the Institute of Geography at the University of Bern, Bern, Switzerland.



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
