# Peer review of "Monthly streamflow forecasting at varying spatial scales in the Rhine basin"

_Hydrology and Earth System Sciences, 2017_

## Referee Comment (RC1) · J. Beckers (Referee) · 12 Jul 2017

General comments

This paper describes the application of the Model Output Statistics (MOS) method to monthly streamflow forecasting in the Rhine River basin. The MOS method is based on empirical relations between predictors (past observations and GCM forecasts) and predictands (monthly mean streamflow) that are obtained through linear regression. Results of several variations of the method using different combinations of predictors are compared in terms of forecast skill.

The paper is well written and clear. In my opinion the paper addresses a relevant scientific topic related to finding methods for streamflow forecasting at the medium

range time scale, which falls within the scope of HESS. Nevertheless, I have a few comments to this paper, as specified below, that I believe should be addressed before I can recommend its publication.

Specific comments

The MOS method is presented as an option for streamflow forecasting at the seasonal time scale (Page1-Line 16, Page 2-Line 18, Page 3-Line 9). However, the results show only forecast skill relative to climatology for the first month ahead. This is usually not what is called seasonal forecasting (rather medium- or extended-range forecasting). So the conclusion must be drawn that no skill was found at the seasonal time scale for any of the models (including ESP and H-TESSEL). This is indeed concluded for the MOS models (Page 17-Line 15), but the suggestion that the performance may be better for particular calendar months (Page 17-Line 15,16) is unfounded and should be removed. Also, the conclusion that H-TESSEL is an interesting option for seasonal (i.e. beyond 1 month lead time) streamflow forecasting (Page 1-Line 13, Page 17-Line 26) is not supported by the results shown in Figure 2 (at least not clear to me).

Moreover, if the MOS method is to be considered for operational streamflow forecasting, it would need to be tested against the more traditional approach, which uses ESP- or GCM-driven hydrologic models. I am not sure which version of H-TESSEL you are using, but since you mention it does not include routing (Page 16-Line 26), this must be a relatively limited model. When comparing the MOS method to this H-TESSEL for zero lead time, the results are not very convincing: according to Table 4 the MAE of the S4* models is worse than for H-TESSEL at Lobith and only marginally better at Basel. Given these results, would the conclusion not be that the MOS methods are not a viable option for operational streamflow forecasting? This conclusion is missing on Page 17.

My final concern with this paper is that it is not clear how the skill of the various models at zero lead time is composed. Probably, the forecast skill for the first 5 days is higher

than for the last 5 days of the 1-month lead time. By averaging over the entire first month this information is lost. It could very well be that the average positive skill for lead times 1-30 days is entirely due to the positive skill for the first few days. Moreover, this positive skill for the first few days may be a result of the persistence of weather patterns in the GCM, similar to that for a normal short range weather forecast.

Related to this is the ESP-revESP analysis. The paper states that there is no clear difference in skill between the ESP and revESP at zero lead time (Page 16-Line 2). But the zero lead time is actually an average for lead times of 1 to 30 days. A separate analysis for lead times of 5, 10, 15, etc days would probably reveal a cross-over from dominance of initial conditions (higher skill of ESP) for short lead times to dominance of meteorological forcing (higher skill of revESP) for longer lead times. But this cannot be seen in the monthly average. Therefore I encourage the authors to do an skill assessment at higher temporal resolution.

Detailed comments

Page 3-Line 23: What approaches do these earlier studies use? Are these (bias-corrected) hydrologic model forecasting studies or do they use MOS/PP?

Page 6-Line 2: I believe the reference for H-TESSEL should be Balsamo et al., 2008 or 2009.

Page 7-Line 6: Sample size is 31? 1981-2011 is 31 years, but you leave out the year of forecast and (according to Section 4.1.3) also the two preceding and subsequent years, so n must be 26.

Page 16-Line 14: It is found that the S4PT model outperforms the ESP model for subcatchments with smooth terrain and weak influence of initial conditions. Can you explain why? I would expect the opposite: the S4PT model (which includes forecast temperature) should do well for catchments that are dominated by snowmelt (rough terrain and strong influence of initial conditions).

---

## Author Comment (AC1) · 14 Jul 2017

**General comments**

The authors thank for the careful review and the clear comments. In general we agree with the comments and think that they bring up some very interesting ideas. We also recognise that we need to use the terms 'option', 'season', and 'operational' more precisely.

**Specific comments**

*The MOS method is presented as an option for streamflow forecasting at the seasonal time scale (Page1-Line 16, Page 2-Line 18, Page 3-Line 9). However, the results show*

[Figure]

*only forecast skill relative to climatology for the first month ahead. This is usually not what is called seasonal forecasting (rather medium- or extended-range forecasting).*

Yes, we agree, the term 'season' needs to be used more carefully, especially when it comes to the actual results of the study (e.g. page 1, line 14).

In the introduction, however, we tried to give a general overview of the approaches that one could apply to integrate GCM output for seasonal (or subseasonal) streamflow forecasting: The traditional model chain GCM - hydrological model; runoff from the land surface component of the GCM run; and PP and MOS based forecast strategies. We do not claim any of these approaches to be superior nor that using GCM output is superior to the ESP approach and its variants. Rather it is an attempt to summarise the existing toolbox, without stating that any particular model is skillful at the (sub)seasonal time scale.

The fact that skill of the tested methods in case of the Rhine basin is restricted to one month ahead forecasts does not invalidate this list – these options still exist (from a technical point of view) and might work or not, depending on the study region, skill of the GCM, validity of the model assumptions, and so on. A forecast at the seasonal time scale remains one, be it skillful or not.

*So the conclusion must be drawn that no skill was found at the seasonal time scale for any of the models (including ESP and H-TESSEL). This is indeed concluded for the MOS models (Page 17-Line 15), but the suggestion that the performance may be better for particular calendar months (Page 17-Line 15,16) is unfounded and should be removed.*

We agree; it was not our purpose to suggest that the models might better perform in particular months, but that the findings are valid only for forecasting all calendar months. Moreover, we should stress that the model might perform better or more worse in particular calendar months, so the results represent somehow an "average" month.

[Figure]

*Also, the conclusion that H-TESSEL is an interesting option for seasonal (i.e. beyond 1 month lead time) streamflow forecasting (Page 1-Line 13, Page 17-Line 26) is not supported by the results shown in Figure 2 (at least not clear to me).*

Yes, we agree, this conclusion is too loosely formulated (again it boils down to the correct usage of the term 'season'). It also implies several arguments that we would like to reformulate here.

The present study tries to test approaches for seasonal streamflow forecasting that have not received much attention in the scientific literature so far, i.e. a MOS approach and runoff simulations from a coupled GCM. The latter simply emerged out of curiosity: While downloading the precipitation and temperature hindcasts from MARS, we realised that runoff is also contained in the list of available parameters. We then decided to request runoff as well and started to use it as a benchmark for the MOS approach. To avoid any misunderstanding: We did not run H-TESSEL at all, we only applied a linear bias correction. Thus, the simulations correspond to the model configuration used within ECMWF's S4 setup.

To our surprise it turned out that these runoff simulations are quite hard to beat, for which reason we included them in the present study (please note that the MOS approach works on an acceptable level when fed with the best available input data, independently of the lead time). Clearly, this finding does only hold for the Rhine basin and H-TESSEL, but nevertheless we think that this point is interesting both from an operational as well as scientific perspective.

Operational, since the runoff simulations are already present after the GCM finishes his run (so no need to run a model by oneself, ECMWF and other institutes already did the job, and notably in a fully coupled way). Scientific, since the land surface model is not aimed to forecast streamflow, but to provide a lower boundary condition for the simulation of the atmosphere. Thus, if we already beat streamflow climatology with runoff from the land surface model, to which extent can we benefit by using models

specifically tailored to forecast streamflow? For example, what are the benefits of including river routing algorithms, groundwater or lake modules, or a higher horizontal resolution? Studies that use GCM output for seasonal streamflow forecasting need to download the necessary parameters anyway, so why not add runoff and use it as a benchmark?

*Moreover, if the MOS method is to be considered for operational streamflow forecasting, it would need to be tested against the more traditional approach, which uses ESP- or GCM-driven hydrologic models.*

Indeed, it would be very interesting to contrast the two tested approaches with the 'traditional' forecast strategy. Since to the best of our knowledge no study exists that we could use for such a comparison, it would be left to us to set up a model and run it with the seasonal climate predictions. This, however, is in our opinion not feasible in the context of the present article, but could lead to a next study.

*I am not sure which version of H-TESSEL you are using, but since you mention it does not include routing (Page 16-Line 26), this must be a relatively limited model. When comparing the MOS method to this H-TESSEL for zero lead time, the results are not very convincing: according to Table 4 the MAE of the S4\* models is worse than for H-TESSEL at Lobith and only marginally better at Basel. Given these results, would the conclusion not be that the MOS methods are not a viable option for operational streamflow forecasting? This conclusion is missing on Page 17.*

Yes, we agree. With respect to the Rhine at Lobith and Basel, the runoff simulations from ECMWF's S4 in combination with a linear bias correction score the same skill as the tested MOS approach does. This needs to be more stressed in the summary as well as in the conclusion. However, we think it is not valid to conclude in general that MOS methods are not a viable option for operational streamflow forecasting. Neither can we speak for the whole MOS family nor for other input data nor other parts of the world. Rather, we argue that MOS in general is part of the "seasonal streamflow

forecasting"-toolbox – with all the disadvantages and advantages of a black box model.

*My final concern with this paper is that it is not clear how the skill of the various models at zero lead time is composed. Probably, the forecast skill for the first 5 days is higher than for the last 5 days of the 1-month lead time. By averaging over the entire first month this information is lost. It could very well be that the average positive skill for lead times 1-30 days is entirely due to the positive skill for the first few days. Moreover, this positive skill for the first few days may be a result of the persistence of weather patterns in the GCM, similar to that for a normal short range weather forecast.*

*Related to this is the ESP-revESP analysis. The paper states that there is no clear difference in skill between the ESP and revESP at zero lead time (Page 16-Line 2). But the zero lead time is actually an average for lead times of 1 to 30 days. A separate analysis for lead times of 5, 10, 15, etc days would probably reveal a cross-over from dominance of initial conditions (higher skill of ESP) for short lead times to dominance of meteorological forcing (higher skill of revESP) for longer lead times. But this cannot be seen in the monthly average. Therefore I encourage the authors to do an skill assessment at higher temporal resolution.*

Again, we agree, this is a very interesting point. If desired, we can run the hindcast for lead times in five day steps. However, we would like to note two arguments to not do so: Basically, it would lead to a new study – our intention was to test a MOS approach with the focus on the spatial scale, not to provide a detailed ESP-revESP analysis for the Rhine basin. The ESP-revESP analysis is used to aid interpretation, and to compare the results with the already existing literature (which is on a monthly basis, too). In addition, we believe that such an experiment mainly tests the assumption of linearity. The present MOS formulation might work to forecast average streamflow of three or even two weeks, but for five days very likely will fail. Therefore, it would be tricky to separate the effects of model misspecification and hydrological persistence.

**Detailed comments**

*Page 3-Line 23: What approaches do these earlier studies use? Are these (bias-corrected) hydrologic model forecasting studies or do they use MOS/PP?*

Yes, this are studies using the GCM - hydrological model chain. We agree, this must be specified.

*Page 6-Line 2: I believe the reference for H-TESSEL should be Balsamo et al., 2008 or 2009.*

Yes, we agree, but suggest to retain the reference to the ECMWF IFS documentation, since the information is taken from there (it also describes the actual H-TESSEL configuration).

*Page 7-Line 6: Sample size is 31? 1981-2011 is 31 years, but you leave out the year of forecast and (according to Section 4.1.3) also the two preceding and subsequent years, so n must be 26.*

Yes, we agree, the sample size equals n=26 for model fitting. We decided to write n=31 since at page 7, the reader only knows that the period amounts to 31 years, but does not know any details about the cross validation. If the article will be accepted for a revision, we will follow your suggestion.

*Page 16-Line 14: It is found that the S4PT model outperforms the ESP model for subcatchments with smooth terrain and weak influence of initial conditions. Can you explain why? I would expect the opposite: the S4PT model (which includes forecast temperature) should do well for catchments that are dominated by snowmelt (rough terrain and strong influence of initial conditions).*

We only can speculate: GCM skill for the Rhine basin is on a low level, and thus hard to detect. When the initial conditions are strongly relevant like in the case of a snow dominated catchment, any error in estimating these initial conditions produces larger errors than the GCM skill can reduce. Thus, we suggest that GCM skill is better detectable in catchments where the relevance of the initial conditions is small. For

example, we would argue that it is hard to successfully force a hydrological model with seasonal climate predictions in a catchment situated in the Alps – if we get the snow pack wrong, the small skill contained in the precipitation and temperature forecasts vanishes.

---

## Author Comment (AC2) · 8 Aug 2017

We would like to take up again the point concerning the resolution of the lead time, since we probably misunderstood the suggestion of J. Beckers in his review.

In the following, we use 'lead time' as the time interval between the release of a forecast and the onset of its validity. For example a mean streamflow forecast for April 21 - 30, produced at March 31, has a lead time of 20 days.

The MOS approach of our study is based on the assumption of linearity and thus needs a certain time window to average the actual predictand. For monthly streamflow averages this assumption seems to be more or less valid. Increasing the temporal resolution of the predictand (e.g. the prediction of 5, 10, 15, ... day mean streamflow

at zero lead time) could be an interesting experiment, but rather to test the assumption of linearity than for a detailed ESP-revESP analysis. This is clearly a disadvantage of using regression instead of a hydrological simulation model.

However (and this was eventually already proposed by J. Beckers) shifting the time window in steps of 5,10,15... days (that is using short lead times) could also reveal some insights. It does not help concerning the ESP-revESP experiment, which remains unresolved at a submonthly time scale, but to detect the skillful time range of the seasonal climate predictions. For example, if the monthly streamflow forecasts based on the seasonal predictions arrive at the MAE of the ESP model at 15 days lead time, we could argue that skill of the seasonal climate predictions is restricted to the first 15 days.

Do you agree with that line of argumentation?

---

## Referee Comment (RC2) · K. Foster (Referee) · 14 Aug 2017

Review - Monthly streamflow forecasting at varying spatial scales in the Rhine basin

Specific comments

As I mentioned in the general comments, Joost has already raised some points especially with regards to the conclusions. I agree with most of his comments so to minimise repetition I will concentrate on other aspects unless where I disagree with him.

Is there a reason why the initial hydrological conditions are not included as predictors (page 3-lines 10-12 and table 2)? Predictors related to storages such as soil moisture content, snow, and reservoir/lake levels all impact future streamflow yet only meteorological predictors are used. I agree that many of these initial storages are affected by the antecedent meteorological conditions, but these connections are not necessarily linear or significant depending on the time frame used. For example, if only predictors for the preceding month are used then there is little connection to snow pack size or reservoir levels and therefore little added value. Thus I miss a description of the time period, and to a lesser extent the domain, for the predictors.

Similarly, I question whether the use of the terms ESP and revESP are technically correct in this paper as it stands. Without any information regarding the initial conditions at the forecast initialisation one can argue that this is not similar to what Wood and Lettenmeier (2008) meant. If it were possible I would suggest the authors include predictors that represented the initial conditions (soil moisture, snow depth, or even streamflow) otherwise they should add a paragraph explaining why the current approach is still an adaption of the VESPA methodology. I believe that the latter may be difficult to justify especially with respect to revESP.

I echo Joost's point where he suggests that the suggestion that the performance may be better for particular months (page 17-lines15, 16) is unfounded as the article stands now. However, I do expect this to be the case and therefore I disagree with him in that this should be removed. Rather I think it would be of interest to include some results or a section that addresses this variability. This can be done in part in the form of a figure along the lines of the one below (figure 1). Related to this, why are the authors concentrating only on the general performance throughout the year? The usefulness of these forecasts may be much higher, even only, during specific times during the year e.x. during the snow melt period or low flow period.

[Figure]

**Figure 1. Forecast skill as a function of lead-time and initialisation date.**

With regards to H-TESSEL, Table 4 shows that it has some skill, at least at the spatial level 1. Have the authors tested using these data as predictors in the MOS approach at levels 2 and 3?

I am unclear as to whether the S4* data is bias corrected. It is now almost common practice for some sort pre-processing or bias correction of the S4* forecast data before use in hydrological forecasting studies and work. The authors note that the quality of seasonal climate predictions for the study area are low (page 3-lines 20,21) but it is not clear to me whether any attempt to bias correct the data, and if I did miss it by what method.

Lastly, the authors mention how the uncertainties in forecasts can be reduced when the quantity of interest is controlled by teleconnection phenomena (page 1-line 17-19). I don't contest that this is true but rather question how it is relevant to the paper because there does not seem to be any more references to such modulation activity or its importance in the rest of the paper.

Technical comments

As mentioned in the general comments I feel that the article is well written so I have two only minor technical comments.

On page 9-line 12 the authors give a secondary citation where I feel that the original citation, or at least inclusion of the original would be strongly advised. Taylor's original article is: *Taylor, K. E. (2001). Summarizing multiple aspects of model performance in a single diagram. Journal of Geophysical Research: Atmospheres, 106(D7), 7183-7192.* The authors are encouraged to check their other sources.

Lastly, there are some minor grammatical errors in the paper; however these do not detract from the readability or arguments made therein. All the same I do suggest that the authors spend a little time to minimise them if time allows.

---

## Author Comment (AC3) · 16 Aug 2017

**General comments**

The authors thank for the careful review and the clear comments. We think that several comments are related to the terminology used in the article, as we often borrow terms from studies with hydrological simulation models, e.g. the 'ESP' and 'revESP' models, 'initial conditions', 'meteorological forcings', and so on. We do not insist to retain these terms in the article and are open for suggestions.

**Specific comments**

*Is there a reason why the initial hydrological conditions are not included as predictors*

[Figure]

*(page 3-lines 10-12 and table 2)? Predictors related to storages such as soil moisture content, snow, and reservoir/lake levels all impact future streamflow yet only meteorological predictors are used. I agree that many of these initial storages are affected by the antecedent meteorological conditions, but these connections are not necessarily linear or significant depending on the time frame used. For example, if only predictors for the preceding month are used then there is little connection to snow pack size or reservoir levels and therefore little added value. Thus I miss a description of the time period, and to a lesser extent the domain, for the predictors.*

We agree, there exist many other potential predictors. We restricted the set of predictors to precipitation and surface air temperature for practical reasons: These variables are available as gridded products, cover the entire study region (and thus are present in all subcatchments), and are available for a long time period; the assumption of independence is more or less valid; and the regression strategy stays simple. As long as it is reasonable to include precipitation and temperature of the target season in the model, then it does so for the 'initial conditions' too. In fact, using this restricted set of predictors guarantees a fair comparison of the predictor combinations and spatial levels as they all rely on the same source of data.

In case of the 'preceding' predictors (the predictors that act as a proxy to catch the initial conditions via the antecedent meteorological conditions), the time aggregation is allowed to vary between 10 and 720 days. The predictors are defined as catchment area averages. Two example plots showing the regression coefficients and aggregation periods of the refRun model at Lobith and Basel (spatial level 1, predictand=30 day mean streamflow, n=100) are at the end of the document. The interpretation of the resulting pattern throughout the year is tricky, but we can see e.g. an increase in the time aggregation of precipitation in spring (Basel level 1, top right plot), i.e. the model tries to catch the snow volume accumulated in winter.

If the article gets accepted for revision, we will clarify this point; if desired we can also include these two figures (but we suggest in the additional materials rather than in the

results section).

*Similarly, I question whether the use of the terms ESP and revESP are technically correct in this paper as it stands. Without any information regarding the initial conditions at the forecast initialisation one can argue that this is not similar to what Wood and Lettenmeier (2008) meant. If it were possible I would suggest the authors include predictors that represented the initial conditions (soil moisture, snow depth, or even streamflow) otherwise they should add a paragraph explaining why the current approach is still an adaption of the VESPA methodology. I believe that the latter may be difficult to justify especially with respect to revESP.*

We agree, the naming of the different predictor combinations is debatable. However, please note that a connection to VESPA does not exist at all (we also do not cite this article). Our intention was to disentangle the predictive power of the 'information prior to the date of prediction' and 'information following the date of prediction'. In our understanding it is exactly this idea that lies at the core of the ESP-revESP framework.

*I echo Joost's point where he suggests that the suggestion that the performance may be better for particular months (page 17-lines15, 16) is unfounded as the article stands now. However, I do expect this to be the case and therefore I disagree with him in that this should be removed. Rather I think it would be of interest to include some results or a section that addresses this variability. This can be done in part in the form of a figure along the lines of the one below (figure 1). Related to this, why are the authors concentrating only on the general performance throughout the year? The usefulness of these forecasts may be much higher, even only, during specific times during the year e.x. during the snow melt period or low flow period.*

We wrote the article with the intention to test the MOS and ECMWF's seasonal climate predictions along several spatial scales, and so decided to ignore variations of predictive skill within the calendar year. We strongly expected that skill of the seasonal climate predictions – if present at all – peaks at a particular spatial level. For example,

the spatial averaging of the fields for Lobith at spatial level 1 might remove important information (i.e. where do we have a climate anomaly?). On the other hand, regressing streamflow of small catchments (say 100 km2) against climate predictions that represent spatial averages of about 5000 km2 could simply amount to a scale mismatch.

We attached a figure following the template of Kean Foster for the S4PT model and the MAE skill score with respect to climatology (however, we did not test for statistical significance). As expected by Kean Foster, skill varies within the calendar year. Maybe we can add a short subsection and discuss this variation of e.g. the S4PT model in the results section. To test for significant deviations from zero we might need to resort to a bootstrap in case the distributional assumptions of the test applied to the complete monthly series are not valid.

*With regards to H-TESSEL, Table 4 shows that it has some skill, at least at the spatial level 1. Have the authors tested using these data as predictors in the MOS approach at levels 2 and 3?*

Yes, we did. The MAE remains virtually the same: 419, 417, 417 m3/s (Lobith levels 1-3) and 191, 186, 184 m3/s (Basel levels 1-3). Please note that runoff from H-TESSEL is also included in the S4Q model (but here in combination with the antecedent meteorological conditions and the time aggregation screening). If desired, we can add these values to Table 4.

*I am unclear as to whether the S4\* data is bias corrected. It is now almost common practice for some sort pre-processing or bias correction of the S4\* forecast data before use in hydrological forecasting studies and work. The authors note that the quality of seasonal climate predictions for the study area are low (page 3-lines 20,21) but it is not clear to me whether any attempt to bias correct the data, and if I did miss it by what method.*

We did not apply any bias correction, since we think it is not useful in case of statistical methods (at least we do not know any study that uses bias corrected predictors for

a regression model). The present formulation of the MOS approaches catches any systematic linear error via the regression coefficients. Obviously, this does not hold for nonlinear systematic errors, but we question that e.g. quantile mapping improves the prediction accuracy, since we work with mean values corresponding to at least 5 days.

The statement about the quality of seasonal climate predictions for the study region should be interpreted from a physical point of view: In the midlatitudes, climate is less dominated by the ENSO, resulting in less skillful seasonal climate predictions when compared to the tropics.

*Lastly, the authors mention how the uncertainties in forecasts can be reduced when the quantity of interest is controlled by teleconnection phenomena (page 1-line 17-19). I don't contest that this is true but rather question how it is relevant to the paper because there does not seem to be any more references to such modulation activity or its importance in the rest of the paper.*

We agree, this statement is not strictly necessary for the article. Rather we tried to sketch the basis for environmental seasonal forecasting in order to start somewhere with the article. Please note that the cited 'slowly-varying and predictable phenomena' are not restricted to the thermal coupling of the oceans and the atmosphere (and potential subsequent teleconnections) – a strong cycle of snow accumulation and subsequent melting or persistence in soil moisture are other examples.

**Technical comments**

*On page 9-line 12 the authors give a secondary citation where I feel that the original citation, or at least inclusion of the original would be strongly advised. Taylor's original article is: Taylor, K. E. (2001). Summarizing multiple aspects of model performance in a single diagram. Journal of Geophysical Research: Atmospheres, 106(D7), 7183-7192. The authors are encouraged to check their other sources.*

We agree, the original citation is more appropriate.

[Figure]

*Lastly, there are some minor grammatical errors in the paper; however these do not detract from the readability or arguments made therein. All the same I do suggest that the authors spend a little time to minimise them if time allows.*

We are not native English speakers, but will try our best.
* * *
[Figure]

**Fig. 1.** Regression coefficients and aggregation time periods of the refRun model at Lobith and spatial level 1 in case of 30 day mean streamflow forecasts..

[Figure]

**Fig. 2.** Regression coefficients and aggregation time periods of the refRun model at Basel and spatial level 1 in case of 30 day mean streamflow forecasts..

[Figure]

**Fig. 3.** MAE skill score with respect to climatology of the S4PT model at different lead times and date of predictions.

---

## Referee Comment (RC3) · J. Beckers (Referee) · 22 Aug 2017

Simon Schick et al. on the point of seasonal forecasting: Yes, we agree, the term 'season' needs to be used more carefully, especially when it comes to the actual results of the study (e.g. page 1, line 14). In the introduction, however, we tried to give a general overview of the approaches that one could apply to integrate GCM output for seasonal (or subseasonal) streamflow forecasting: The traditional model chain GCM - hydrological model; runoff from the land surface component of the GCM run; and PP and MOS based forecast strategies. We do not claim any of these approaches to be superior nor that using GCM output is superior to the ESP approach and its variants. Rather it is an attempt to summarise the existing toolbox, without stating that any particular model is skillful at the (sub)seasonal time scale. The fact that skill of the

tested methods in case of the Rhine basin is restricted to one month ahead forecasts does not invalidate this list – these options still exist (from a technical point of view) and might work or not, depending on the study region, skill of the GCM, validity of the model assumptions, and so on. A forecast at the seasonal time scale remains one, be it skillful or not.

Response from Joost Beckers: Yes but using the term seasonal forecasting creates expectations. Since no skill is found beyond the one-month lead time, I would recommend to keep references to seasonal forecasting minimal.

Simon Schick et al. on the suggestion that the performance of MOS may be better for particular calendar months: We agree; it was not our purpose to suggest that the models might better perform in particular months, but that the findings are valid only for forecasting all calendar months. Moreover, we should stress that the model might perform better or more worse in particular calendar months, so the results represent somehow an "average"month.

Response from Joost Beckers: I agree with the remark by K. Foster that it would be even better to demonstrate this variation in performance in an additional figure or table.

Simon Schick et al. on the limited skill of H-TESSEL at the seasonal time scale: Yes, we agree, this conclusion is too loosely formulated (again it boils down to the correct usage of the term 'season'). It also implies several arguments that we would like to reformulate here. The present study tries to test approaches for seasonal streamflow forecasting that have not received much attention in the scientific literature so far, i.e. a MOS approach and runoff simulations from a coupled GCM. The latter simply emerged out of curiosity: While downloading the precipitation and temperature hindcasts from MARS, we realised that runoff is also contained in the list of available parameters. We then decided to request runoff as well and started to use it as a benchmark for the MOS approach. To avoid any misunderstanding: We did not run H-TESSEL at all, we only applied a linear bias correction. Thus, the simulations correspond to the model

configuration used within ECMWF's S4 setup. To our surprise it turned out that these runoff simulations are quite hard to beat, for which reason we included them in the present study (please note that the MOS approach works on an acceptable level when fed with the best available input data, independently of the lead time). Clearly, this finding does only hold for the Rhine basin and H-TESSEL, but nevertheless we think that this point is interesting both from an operational as well as scientific perspective. Operational, since the runoff simulations are already present after the GCM finishes his run (so no need to run a model by oneself, ECMWF and other institutes already did the job, and notably in a fully coupled way). Scientific, since the land surface model is not aimed to forecast streamflow, but to provide a lower boundary condition for the simulation of the atmosphere. Thus, if we already beat streamflow climatology with runoff from the land surface model, to which extent can we benefit by using models specifically tailored to forecast streamflow? For example, what are the benefits of including river routing algorithms, groundwater or lake modules, or a higher horizontal resolution? Studies that use GCM output for seasonal streamflow forecasting need to download the necessary parameters anyway, so why not add runoff and use it as a benchmark?

Response from Joost Beckers: Thanks for that elaborate answer. Indeed the H-TESSEL is a excellent benchmark and it was good to include it in the analysis.

Simon Schick et al. on the testing of MOS against ESP- or GCM-driven hydrologic models: Indeed, it would be very interesting to contrast the two tested approaches with the 'traditional' forecast strategy. Since to the best of our knowledge no study exists that we could use for such a comparison, it would be left to us to set up a model and run it with the seasonal climate predictions. This, however, is in our opinion not feasible in the context of the present article, but could lead to a next study.

Response from Joost Beckers I understand that this would go beyond the scope of your project. Note, though, that there are organizations that have such Rhine models running operationally (e.g. BfG, Germany and Rijkswaterstaat, Netherlands).
Simon Schick et al. on the version of H-TESSEL: Yes, we agree. With respect to the Rhine at Lobith and Basel, the runoff simulations from ECMWF's S4 in combination with a linear bias correction score the same skill as the tested MOS approach does. This needs to be more stressed in the summary as well as in the conclusion. However, we think it is not valid to conclude in general that MOS methods are not a viable option for operational streamflow forecasting. Neither can we speak for the whole MOS family nor for other input data nor other parts of the world. Rather, we argue that MOS in general is part of the "seasonal streamflow forecasting"-toolbox – with all the disadvantages and advantages of a black box model.

Response from Joost Beckers: I agree that a general conclusion about the viability of any of the methods would go too far. Your last sentence puts it in the right perspective.

Simon Schick et al. on the point of resolution and ESP-revESP: We would like to take up again the point concerning the resolution of the lead time, since we probably misunderstood the suggestion of J. Beckers in his review. In the following, we use 'lead time' as the time interval between the release of a forecast and the onset of its validity. For example a mean streamflow forecast for April 21 - 30, produced at March 31, has a lead time of 20 days. The MOS approach of our study is based on the assumption of linearity and thus needs a certain time window to average the actual predictand. For monthly streamflow averages this assumption seems to be more or less valid. Increasing the temporal resolution of the predictand (e.g. the prediction of 5, 10, 15, ... day mean streamflow at zero lead time) could be an interesting experiment, but rather to test the assumption of linearity than for a detailed ESP-revESP analysis. This is clearly a disadvantage of using regression instead of a hydrological simulation model. However (and this was eventually already proposed by J. Beckers) shifting the time window in steps of 5,10,15... days (that is using short lead times) could also reveal some insights. It does not help concerning the ESP-revESP experiment, which remains unresolved at a submonthly time scale, but to detect the skillful time range of the seasonal climate predictions. For example, if the monthly streamflow forecasts
based on the seasonal predictions arrive at the MAE of the ESP model at 15 days lead time, we could argue that skill of the seasonal climate predictions is restricted to the first 15 days. Do you agree with that line of argumentation?

Response from Joost Beckers: Yes, I agree. Your proposed investigation would reveal how the skill at zero lead time decays to the ESP value at some longer lead time. If it the ESP value is already reached after one week, then the conclusion must be that the skill at zero lead time is entirely due to a good prediction of the flow for the first few days after forecast time. Possibly, you can even reconstruct the MAE for these first few days from the monthly flow MAEs at zero and 1-week lead times. However, this would be a somewhat indirect way of determining the skill at higher temporal resolution. I do not understand why a more direct analysis of the skill for the first week is not possible. Why does the MOS method require a flow averaging over a monthly interval? If you would apply the method to weekly or even daily flows, surely the results will become more noisy, but on average the linear relationships would still hold, or not? If such an analysis at a shorter interval would be possible, this would also enable to investigate the crossover from ESP to revESP in more detail.

---

## Author Comment (AC4) · 25 Aug 2017

Thank you very much for taking time for a second comment. We are currently working on a final author response where we try to compile your and Kean Foster's suggestions, such that the editors can decide whether the article gets accepted for a revision.

---

## Author Comment (AC5) · 26 Aug 2017

The authors thanks for the good and comprehensive reviews by Joost Beckers and Kean Foster, which reveal some important deficiencies of our article and hint some interesting ideas. Please find below a list of how we plan to work the article over if it gets accepted for a revision. Since we agree with the detailed/technical comments of both reviewers, we include here only the major issues.

**Terminology**

- We check the article for the correct usage of the terms 'season' and 'operational' in order to avoid wrong expectations. It must be clearly stressed that – on average

[Figure]

– the prediction skill of the MOS approach is restricted to one month ahead.

- We will revise the naming of the different predictor combinations, e.g. preMet (preceding meteorology) and subMet (subsequent meteorology) instead of ESP and revESP (up to now, we do not have a better idea). However, we suggest to retain the link to the ESP-revESP framework for the motivation of such a preMet-subMet analysis (or whatever its name), since we originally aimed to mimic the ESP-revESP framework in a statistical context, and the underlying idea is not at all ours.

**Introduction**

- As pointed out by Kean Foster, we need to clarify the first paragraph about the 'slowly-varying and predictable phenomena' as well as teleconnections and why they are relevant for our study (page 1, line 16). Also we should mention why it is hard to make seasonal climate predictions for Europe compared to the tropics (page 3, line 19).

**Results**

- We add plots (and corresponding paragraphs) similar to the ones attached to our response of Kean Foster's review, which show the variation of the regression coefficients and time aggregation periods (maybe it can help to make the black box model a little bit less black) as well as the variation of skill versus the date of prediction. We suggest to put the former to the additional materials, while the latter is in our opinion appropriate for the results section.

- We add the MAE of the bias corrected H-TESSEL runoff for spatial levels 2 and 3 to table 4.

- Concerning the suggestion of Joost Beckers (variation of the predictand's time aggregation at zero lead time, e.g. streamflow averages of 1,3,5,10,15,...,30 days) we propose one of the following:

  1. We include such an analysis in the article, but drop the results about the comparison of skill versus geographic attributes (i.e. figure 4 and the corresponding paragraphs). We have the impression that otherwise the article gets too lengthy.
  2. We do not include such an analysis and stick with the 'skill versus geographic attributes' results, but instead discuss such an experiment in the discussion and conclusion section for further research.

We have not yet tested the MOS approach at such small time average windows, so we have no clue about the direction of possible results. As we often read that catchments are complex, non-linear systems, we silently assumed that a linear regression model fails at such small forecast windows. On the other hand, studies exist in which ARIMA models are fitted to daily streamflows, so who knows?

As already noted in our response to Joost Beckers' review, we think that such a preMet-subMet analysis for short time intervals would be an interesting experiment. However, there is simply no guarantee that the MOS approach does not crash (at least we can verify this point with the refRun model, which has access to the best available input data).

Maybe the editor can provide some guidance regarding this point.

**Discussion**

- We add a section about pros and cons of MOS based streamflow forecasts, e.g. that we get a bias correction for free (in case of our article restricted to unconditional biases and conditional biases that are linear), that it is not possible to make predictions at ungauged sites, and so forth.

- If the reviewers and the editor agree, we could add some MAE or MSE estimates of deterministic hydrological models for the Rhine at Basel or Lobith. Up to now, we do not have any relevant reference from the scientific literature, but eventually we could find one or two references since climate change impact studies often report MAE, MSE, NSE and friends on a monthly or yearly basis. Clearly, such a comparison would mix the contexts of climate change projections and S2S forecasting, but could provide a rough order of magnitude.

- We need to stress that by using meteorological input variables only (aside from the S4Q model), we end up with a loose and fuzzy proxy for the initial conditions. Additional input variables most probably improve the prediction accuracy, but impose technical and practical disadvantages that we aim to avoid for the present study (inconsistent data availability, predictor dependence).

**Conclusion**

- We should more elaborate on why we think that the performance of the H-TESSEL benchmark is interesting: H-TESSEL is skillful against climatology and the MOS approaches in case of the Rhine basin, though H-TESSEL runoff (within the S4 system) is not intended to provide streamflow predictions. At this point, we think it is legitimate to have an outlook regarding operational applications: If

the setup of a hydrological model or the subsequent production of streamflow forecasts is not feasible, but streamflow observations are available, then a simple bias correction of H-TESSEL runoff from the S4 system could be worth to test.

---

## Author Response (AR1)

**Contents**

**1 Response**

**1.1 Review by Joost Beckers**

*The MOS method is presented as an option for streamflow forecasting at the seasonal time scale (Page 1-Line 16, Page 2-Line 18, Page 3-Line 9). However, the results show only forecast skill relative to climatology for the first month ahead. This is usually not what is called seasonal forecasting (rather medium- or extended-range forecasting). So the conclusion must be drawn that no skill was found at the seasonal time scale for any of the models (including ESP and H-TESSEL). This is indeed concluded for the MOS models (Page 17-Line 15), but the suggestion that the performance may be better for particular calendar months (Page 17-Line 15,16) is unfounded and should be removed. Also, the conclusion that H-TESSEL is an interesting option for seasonal (i.e. beyond 1 month lead time) streamflow forecasting (Page 1-Line 13, Page 17-Line 26) is not supported by the results shown in Figure 2 (at least not clear to me).*

Wherever possible we removed the term 'season'. Its usage is now restricted to the 'seasonal climate predictions' of ECMWF and to paragraphs dealing with seasonal forecasting in general. In addition, we added a short subsection to the results that looks at the variation of forecast skill within the calendar year (this was also proposed by Kean Foster).

*Moreover, if the MOS method is to be considered for operational streamflow forecasting, it would need to be tested against the more traditional approach, which uses ESP- or GCM-driven hydrologic models.*

We removed the therm 'operational' from the article; it remains in the introduction (ESP approach as the de facto standard in operational seasonal streamflow forecasting) and in the 'operational analysis' of numerical weather forecasting.

*I am not sure which version of H-TESSEL you are using, but since you mention it does not include routing (Page 16-Line 26), this must be a relatively limited model. When comparing the MOS method to this H-TESSEL for zero lead time, the results are not very convincing: according to Table 4 the MAE*
5 *of the S4\* models is worse than for H-TESSEL at Lobith and only marginally better at Basel. Given these results, would the conclusion not be that the MOS methods are not a viable option for operational streamflow forecasting? This conclusion is missing on Page 17.*

We agree, it seems that the simplest approach (linear bias correction of H-
10 TESSEL runoff) often performs best. This point is now more stressed in the conclusion.

*My final concern with this paper is that it is not clear how the skill of the various models at zero lead time is composed. Probably, the forecast skill for the first 5 days is higher than for the last 5 days of the 1-month lead time.*
15 *By averaging over the entire first month this information is lost. It could very well be that the average positive skill for lead times 1-30 days is entirely due to the positive skill for the first few days. Moreover, this positive skill for the first few days may be a result of the persistence of weather patterns in the GCM, similar to that for a normal short range weather forecast.*
20 *Related to this is the ESP-revESP analysis. The paper states that there is no clear difference in skill between the ESP and revESP at zero lead time (Page 16-Line 2). But the zero lead time is actually an average for lead times of 1 to 30 days. A separate analysis for lead times of 5, 10, 15, etc days would probably reveal a cross-over from dominance of initial conditions*
25 *(higher skill of ESP) for short lead times to dominance of meteorological forcing (higher skill of revESP) for longer lead times. But this cannot be seen in the monthly average. Therefore I encourage the authors to do an skill assessment at higher temporal resolution.*

We did an experiment similar to the monthly analysis for five day mean
30 streamflow and lead times of $0, 5, \ldots, 175$ days.

*Page 3-Line 23: What approaches do these earlier studies use? Are these (bias-corrected) hydrologic model forecasting studies or do they use MOS/PP?*

We tried to clarify the paragraph – these are all studies using hydrological models forced by subseasonal or seasonal climate predictions.

35 *Page 6-Line 2: I believe the reference for H-TESSEL should be Balsamo et al., 2008 or 2009.*

We added Balsamo et al., 2009 to the references.

*Page 7-Line 6: Sample size is 31? 1981-2011 is 31 years, but you leave out the year of forecast and (according to Section 4.1.3) also the two preceding and subsequent years, so n must be 26.*

We equated $n$ to 26.

*Page 16-Line 14: It is found that the S4PT model outperforms the ESP model for subcatchments with smooth terrain and weak influence of initial conditions. Can you explain why? I would expect the opposite: the S4PT model (which includes forecast temperature) should do well for catchments that are dominated by snowmelt (rough terrain and strong influence of initial conditions).*

We only can speculate: GCM skill for the Rhine basin is on a low level, and thus hard to detect. When the initial conditions are strongly relevant like in the case of a snow dominated catchment, any error in estimating these initial conditions produces larger errors than the GCM skill can reduce. Thus, we suggest that GCM skill is better detectable in catchments where the relevance of the initial conditions is small. For example, we would argue that it is hard to successfully force a hydrological model with seasonal climate predictions in a catchment situated in the Alps – if we get the snow pack wrong, the small skill contained in the precipitation and temperature forecasts vanishes. This point is now included in the discussion.

**1.2   Review by Kean Foster**

*Is there a reason why the initial hydrological conditions are not included as predictors (page 3-lines 10-12 and table 2)? Predictors related to storages such as soil moisture content, snow, and reservoir/lake levels all impact future streamflow yet only meteorological predictors are used. I agree that many of these initial storages are affected by the antecedent meteorological conditions, but these connections are not necessarily linear or significant depending on the time frame used. For example, if only predictors for the preceding month are used then there is little connection to snow pack size or reservoir levels and therefore little added value. Thus I miss a description of the time period, and to a lesser extent the domain, for the predictors.*

We agree, there exist many other potential predictors. We restricted the set of predictors to precipitation and surface air temperature for practical reasons: These variables are available as gridded products, cover the entire

study region (and thus are present in all subcatchments), and are available for a long time period; the assumption of independence is more or less valid; and the regression strategy stays simple. As long as it is reasonable to include precipitation and temperature of the target season in the model, then it does so for the 'initial conditions' too. In fact, using this restricted set of predictors guarantees a fair comparison of the predictor combinations and spatial levels as they all rely on the same source of data.

In case of the 'preceding' predictors (the predictors that act as a proxy to catch the initial conditions via the antecedent meteorological conditions), the time aggregation is allowed to vary between 10 and 720 days. The predictors are defined as catchment area averages. Two example plots showing the regression coefficients and aggregation periods of the refRun model at Lobith and Basel are now included in the additional materials.

*Similarly, I question whether the use of the terms ESP and revESP are technically correct in this paper as it stands. Without any information regarding the initial conditions at the forecast initialisation one can argue that this is not similar to what Wood and Lettenmeier (2008) meant. If it were possible I would suggest the authors include predictors that represented the initial conditions (soil moisture, snow depth, or even streamflow) otherwise they should add a paragraph explaining why the current approach is still an adaption of the VESPA methodology. I believe that the latter may be difficult to justify especially with respect to revESP.*

We renamed the ESP to preMet and revESP to subMet.

*I echo Joost's point where he suggests that the suggestion that the performance may be better for particular months (page 17-lines15, 16) is unfounded as the article stands now. However, I do expect this to be the case and therefore I disagree with him in that this should be removed. Rather I think it would be of interest to include some results or a section that addresses this variability. This can be done in part in the form of a figure along the lines of the one below (figure 1). Related to this, why are the authors concentrating only on the general performance throughout the year? The usefulness of these forecasts may be much higher, even only, during specific times during the year e.x. during the snow melt period or low flow period.*

We added a subsection to the results that looks at the variation of forecast skill within the calendar year.

*With regards to H-TESSEL, Table 4 shows that it has some skill, at least at the spatial level 1. Have the authors tested using these data as predictors in the MOS approach at levels 2 and 3?*

We completed Tab. 4 with the corresponding values.

*I am unclear as to whether the S4\* data is bias corrected. It is now almost common practice for some sort pre-processing or bias correction of the S4\* forecast data before use in hydrological forecasting studies and work. The authors note that the quality of seasonal climate predictions for the study area are low (page 3-lines 20,21) but it is not clear to me whether any attempt to bias correct the data, and if I did miss it by what method.*

We did not apply any bias correction, since we think it is not useful in case of statistical methods (at least we do not know any study that uses bias corrected predictors for a regression model). The present formulation of the MOS approaches catches any systematic linear error via the regression coefficients. Obviously, this does not hold for nonlinear systematic errors, but we question that e.g. quantile mapping improves the prediction accuracy, since we work with mean values corresponding to at least 5 days.

*Lastly, the authors mention how the uncertainties in forecasts can be reduced when the quantity of interest is controlled by teleconnection phenomena (page 1-line 17-19). I don't contest that this is true but rather question how it is relevant to the paper because there does not seem to be any more references to such modulation activity or its importance in the rest of the paper.*

We agree, this statement is not strictly necessary for the article. Rather we tried to sketch the basis for environmental seasonal forecasting in order to start somewhere with the article. Please note that the cited 'slowly-varying and predictable phenomena' are not restricted to the thermal coupling of the oceans and the atmosphere (and potential subsequent teleconnections) – a strong cycle of snow accumulation and subsequent melting or persistence in soil moisture are other examples. We tried to clarify the corresponding paragraph.

*On page 9-line 12 the authors give a secondary citation where I feel that the original citation, or at least inclusion of the original would be strongly advised. Taylor's original article is: Taylor, K. E. (2001). Summarizing multiple aspects of model performance in a single diagram. Journal of Geophysical Research: Atmospheres, 106(D7), 7183-7192. The authors are encouraged to check their other sources.*

We added Taylor, 2001 to the references. Otherwise, there are only minor changes in the list of references. The book published by the National Academies ('Next Generation Earth System Prediction') obviously is grey

literature – however, we retained it in the article since we think it is a good book: It summarises the state-of-the art in research and industry, it looks at seasonal forecasting from a broader perspective (though heavily biased towards climate predictions), and it is written and reviewed by well-known experts in the field.

*Lastly, there are some minor grammatical errors in the paper; however these do not detract from the readability or arguments made therein. All the same I do suggest that the authors spend a little time to minimise them if time allows.*

We tried our best (obviously, we are not native English speakers). However, we also trust the copy-editing skills of the Copernicus team to remove the remaining errors, if the manuscript gets considered for publication.

**1.3   Comments by Fredrik Wetterhall**

*Regarding the analysis of the time aggregation, I would suggest that you add that to the paper since it is worth testing it. If the article gets too lengthy you can remove the skill vs geographical attributes to supplementary material. However, I do not feel that the paper is too long.*

We added the experiment for the five day mean streamflow to the results and retained the skill vs. geographical attributes results.

*I am not sure that adding MAE or MSE results from the literature would add anything to this particular study, so I would not recommend that.*

We followed your recommendation.

*Please also do a language check. I would in particular suggest to not use GCM, especially in the terminology of weather forecasting. The term I would suggest here is NWP (Numerical Weather Prediction) or even ESM (Earth System Model). The term GCM is too broad, for example it does not explicitly include the analysis, which is an essential part of weather forecasting.*

We tried to consistently use the term earth system model. 'GCM' remains in the text for a few exceptions, i.e. 'atmospheric GCM' or 'coupled atmosphere-ocean-land GCMs'.

**2  Output of LaTeXdiff**

Below you find the output of LaTeXdiff. Since we had to rewrite the code to gain some speed and we also rerun the complete experiment, there are some minor changes in the results (e.g. MAE values) – these changes are introduced by the way the sequence of random integers is drawn in order to generate the bootstrap replicates. However, these changes neither affect the discussion of the results nor the conclusions.

Please excuse the layout of Table 5 – we have no clue why LaTeXdiff fails. Eventually the table contains improper tex code, however, we could not solve this issue.

[revised manuscript text omitted]